# Adipose Tissue Aging and Metabolic Disorder, and the Impact of Nutritional Interventions

**DOI:** 10.3390/nu14153134

**Published:** 2022-07-29

**Authors:** Xiujuan Wang, Meihong Xu, Yong Li

**Affiliations:** Department of Nutrition and Food Hygiene, School of Public Health, Peking University, Beijing 100191, China; 1510306116@pku.edu.cn (X.W.); xumeihong@bjmu.edu.cn (M.X.)

**Keywords:** adipose tissue, aging, caloric restriction, vitamin, resveratrol

## Abstract

Adipose tissue is the largest and most active endocrine organ, involved in regulating energy balance, glucose and lipid homeostasis and immune function. Adipose tissue aging processes are associated with brown adipose tissue whitening, white adipose tissue redistribution and ectopic deposition, resulting in an increase in age-related inflammatory factors, which then trigger a variety of metabolic syndromes, including diabetes and hyperlipidemia. Metabolic syndrome, in turn, is associated with increased inflammatory factors, all-cause mortality and cognitive impairment. There is a growing interest in the role of nutritional interventions in adipose tissue aging. Nowadays, research has confirmed that nutritional interventions, involving caloric restriction and the use of vitamins, resveratrol and other active substances, are effective in managing adipose tissue aging’s adverse effects, such as obesity. In this review we summarized age-related physiological characteristics of adipose tissue, and focused on what nutritional interventions can do in improving the retrogradation and how they do this.

## 1. Introduction

Aging is defined as decline in the biological process of self-organizing systems and reduction in ability to adapt to the environment, combined with multiple biochemical processes at different levels [1]. Normal aging is accompanied by body composition changes, including having less muscle mass and more fat mass. Adipose tissue is the largest energy reservoir and endocrine organ in the body, participating in regulation of the energy balance, glucolipid homeostasis and immune function [2]. Adipose tissue aging is one of the common features in aging, and obesity plays a key role in the mechanisms and pathways of longevity, age-related diseases, inflammation and metabolic dysfunction [2,3].

The adipose tissue aging process is associated with brown adipose tissue whitening, white adipose tissue redistribution and ectopic deposition. The senescence of adipocytes causing an increase in age-related inflammation factors result in various metabolic syndromes, including diabetes, hyperlipidemia, cardiovascular issues, cognitive impairment and functional decline. In turn, the metabolic syndrome disorder causes the high level of body inflammation. Therefore, adipose tissue is the link between age-related inflammation and metabolic dysfunction.

There is a growing interest in the effect nutritional interventions can have on adipose tissue aging. Nutritional interventions, including caloric restriction and the extra supplementation of biochemical substances, have the advantages of being safe, non-invasive, low-cost and easy to access. To highlight and clarify the benefits of nutritional intervention, we summarized the dynamic biological process of age-related adipose tissue, and focused on how nutrients can help improve adipose tissue function.

## 2. Aging of Adipose Tissue

### 2.1. White Adipose Tissue Aging

White adipose tissue (WAT) is mainly distributed in subcutaneous tissues, omentum and around viscera. WAT consists of a large number of adipose cells and other cells, connective tissue matrix, blood vessels and nerve tissues. Non-adipocyte cell components include inflammatory cells (macrophages), immune cells, pre-cells, and fibroblasts [4]. A white adipocyte has a large lipid droplet in the center, which makes up about 90% of the cell volume. WAT stores nutrients in the form of triglycerides and plays an important role in buffering the supply and demand of nutrients by preventing toxic build-up of excess nutrients in non-adipose tissues. White adipocytes can change size quickly and widely in response to widespread changes in nutrient availability [3,5]. WAT is also the largest endocrine organ. It secretes a variety of adipokines, such as leptin and adiponectin, to communicate with other metabolic-related organs and participate in the regulation of nutrient metabolism [5].

Adipose tissue increases in middle age and decreases when people get old [3,6,7], and is accompanied by fat being redistributed to different fat depots, that is, from the subcutaneous to the abdominal visceral depots [3,8]. There is evidence that pre-adipocytes from subcutaneous and visceral depots actually belong to different cell subtypes. Regional variations in adipose tissue size and function result from local variations in adipocyte composition, circulation, nerves, and other factors [9]. Epidemiological studies have shown that visceral adipose tissue accumulation is a risk factor for age-related metabolic diseases, and subcutaneous adipose tissue is protective. Insulin resistance and glucose metabolism disorders are more closely related to regional adipose tissue distribution than total fat mass [10].

Subcutaneous adipose tissue (SAT) is the largest reservoir of adipose tissue and a prime place to store excess fat in mammals. Early in life, SAT is capable of rapidly altering its endocrine, inflammatory, and metabolic functions and is highly adaptable to changes in the internal environment caused by nutrient deficiency, cold exposure, or infection [11]. SAT stores excess lipids in the form of expanding adipocytes and recruits pre-adipocytes to differentiate into subcutaneous adipocytes [12]. In the process of aging or obesity, storage capacity of subcutaneous fat depots tend to be limited by the expansion ability. Excessive lipids start to be stored in other depots that are harmful for health, including around visceral [13], myocardial and skeletal muscle, liver, and pancreas [12]. Ectopic adipose tissue leads to outside lipids building up toxicity in tissues, causing insulin resistance and dyslipidemia [5,14]. The limited expansion and storage capacity of SAT has proved to be the main driver of obesity-related metabolic complications [15]. Kim SM [13] traced fat in WAT by an isotope labeling method, and found that the main mechanism of adult obesity is adipocyte hypertrophy. The aging process is accompanied by loss of SAT plasticity, which is closely related to insulin resistance. Impaired SAT plasticity can lead to insulin resistance in mice, and good SAT plasticity effectively can prevent age-related metabolic dysfunction in mice, including insulin resistance and impaired thyroid axis function [16]. Transgenic mice with high adiponectin expression showed pathological increases in SAT, but insulin sensitivity consistent with that of skinny mice [17].

Visceral adipose tissue (VAT) accounts for about 10 to 20 percent of total fat mass in men and 5 to 8 percent in women. VAT tends to increase with age [4], which may be related to age-related loss of SAT adipocyte plasticity and insufficient supply of pre-adipocytes [11,15]. Postmenopausal women and elderly men both show significant decrease in percentage of leg fat and increase in central fat [18,19], but the increase of VAT has a negative impact on skeletal muscle mass [20]. Compared with SAT, VAT has larger adipocyte diameter and lower insulin sensitivity [15]. VAT could be more prone to fatty acid overflow from lipoprotein hydrolysis than SAT due to its low insulin sensitivity. These free fatty acids reach the liver via the portal vein, resulting in liver exposure to a higher concentration of free fatty acids than the periphery [9]. At present, VAT is generally considered to be a major risk factor for insulin resistance, type II diabetes, cardiovascular disease, stroke, metabolic syndrome and death [21,22,23]. Hypertriglyceridemia, increased FFA in blood, increased inflammatory factors, liver insulin resistance and inflammation, increased hepatic VLDL synthesis and secretion, decreased clearance rate of triglyceride-rich lipoprotein, small and dense LDL particles, decreased HDL level and other age-related metabolic changes are all related to VAT accumulation [24]. In recent years, other studies have found that central obesity caused by VAT accumulation is associated with mild cognitive impairment in the elderly [25,26], and high VAT has negative effects on cerebral subcortical gray matter volume, hippocampal volume and memory [27]. High VAT is related to the aging of the adult brain and has a significantly negative correlation with total brain volume [28]. In addition, VAT is the main source of BMP in aging mice, which can induce senescence of cardiac fibroblasts and lead to changes in cardiac structure and function [29]. VAT also has a significantly negative impact on the respiratory system function of the elderly [30]. The inflammation level of VAT is associated with age-related structural changes in the mouse brain, and VAT removal significantly improved the permeability of the blood brain barrier in mice, contributing to the recovery of ischemic brain injury in mice [31].

### 2.2. Brown Adipose Tissue in Aging

Brown adipose tissue (BAT) is composed of brown adipocytes. BAT is rich in capillaries, and the brown adipocytes are scattered with many small fat droplets. The mitochondria are abundant, causing BAT to appear to be brown. Brown adipocytes of rodents were distributed in defined BAT depots, the largest of which were located in interscapular BAT depots. There are also some small BAT depots distributed around the kidney and aorta [32]. Similar to mice, a large number of active brown adipocytes are located in neonatal interscapular and perirenal BAT depots to help maintain core body temperature. It used to be thought that there was no BAT with metabolic activity in adults. Nowadays, with the development of imaging technology, PET-CT scanning found that active BAT also exists in adults, which is mainly distributed in the neck, supraclavicular, paraspinal, mediastinum, surrounding aorta and kidney [33]. BAT is regarded as a new target for the treatment of obesity and obesity-related metabolic disorders, due to its ability to consume excess nutrients in the body.

Although both white and brown adipocytes store triglycerides, they differ greatly in molecular and physiological parameters, involving lipid droplet conformation, mitochondrial content and activity, and nuclear localization [34]. Brown adipocytes uniquely express uncoupling protein 1 (UCP1) in the inner membrane of mitochondria. Under the stimulation of cold and norepinephrine released from sympathetic nerve endings, intracellular cAMP level increases, thus activating PKA and UCP1 and increasing UCP1 expression. UCP1 provides a low-impedance pathway for protons to return directly to the interstitium, disrupt the proton gradient, bypass ATP synthase complex, and induce mitochondrial respiratory uncoupling, thus, completing non-shivering thermogenesis [35]. Like WAT, BAT also participates in the regulation of nutrient metabolism by secreting Nrg 4, IGF1, IL-6, FGF21, VEGF A/B, adiponectin, visfatin and other bioactive substances [36]. Studies confirmed that the use of cold stimulation to increase BAT volume has a positive effect on lipid metabolism in adults, and BAT volume is significantly correlated with adipose decomposition, FFA circulation, FFA oxidation and increased insulin sensitivity of adipose tissue [37]. Activated BAT plays an important role in energy consumption, glucose balance and human insulin sensitivity, suggesting that BAT may be useful as an anti-diabetic tissue in the human body [38].

The signs of BAT aging include volume atrophy, decreased mitochondrial function and reduced UCP1 activation [35]. As age progresses, macrophage infiltration and excess triglyceride accumulation promotes whitening within the BAT sample phenotype, and gradually the brown adipocytes are replaced [39,40]. The differentiation of BAT precursor cells in elderly mice was impaired [41], and brown adipocytes showed a decreased number of intracellular lipid droplets, enlarged diameter and enlarged cells. Correspondingly, the level of WAT browning decreased and beige fat decreased [42]. Aging brown adipocytes lose their proliferation ability and UCP1 expression ability in response to cold stimulation. Delaying the BAT whitening in elderly mice and maintaining the BAT morphological function can prevent metabolic diseases related to age and obesity [40]. BAT can also be used as a target to improve age-related fat mass gain [41].

## 3. Dynamic Physiological Process in Adipose Tissue Aging

### 3.1. Inflammation

Systemic chronic inflammatory response is one of the most important characteristics of aging [43]. Adipose tissue is often considered to be the main source of systemic inflammatory factors [44]. Aging results in adipose tissue dysfunction, including reduced adipose progenitor cell function, increased secretion of inflammatory cytokines and chemokines, immune cell infiltration, increased senescent cell and senescence associated secretory phenotype (SASP). SASP refers to the enhancement of various cytokines secreted by senescent cells. SASP cytokines include chemokines, inflammatory cytokines, growth factors and matrix metalloproteinases and other aging information transfer secretions. Under normal circumstances, increased SASP can recruit and activate immune cells around senescent cells, promoting the elimination of senescent cells by the immune system. However, with age, aging adipocytes gradually accumulate, and SASP continues to secrete, which induces insulin resistance and systemic chronic inflammation by promoting the inflammatory process of pre-adipocytes, inhibiting differentiation and driving immune cell infiltration, causing systemic metabolic disorders and functional decline, ultimately leading to diabetes and weakness [11,45].

The mechanism of WAT pathogenic inflammation focuses on the pro-inflammatory pathway (including JNK and NF-κB and other pathways) activating the response of adipocytes to metabolic stress caused by obesity. This metabolic stress leads to the change of WAT secretion spectrum, that is, an increase in the release of pro-inflammatory cytokines (TNF-α) and reduction in the production of anti-inflammatory cytokines and adipokines (adiponectin) [46]. Inflammatory factors are an important part of SASP. IL-6 and TNF-α have been shown to rise with age [3]. Compared with SAT, VAT is more likely to increase the level of inflammation, and high VAT can significantly increase TNF-α, IL-6, CRP, MCP-1 levels [4].

Macrophages differentiate into M1 and M2 subsets under different stimuli. Among these, M1 cells secrete pro-inflammatory factors and are strongly destructive. Obese or aging adipose tissue shows higher macrophage accumulation, and macrophages are more likely to polarize into inflammatory M1 subsets [47,48]. These macrophages surround dead or dying adipocytes and form a typical crown-like structure, which can be quantified as a means of assessing the inflammation level in adipose tissue [49,50]. NK cells participate in the process of macrophage polarization. In VAT, fat stress caused by obesity causes the up regulation of NCR1 ligand on adipocytes, activates NK cells in vivo, promotes their proliferation and induces the production of IFN-γ. When NK cell activation is inhibited, obesity induced M1 macrophage polarization is significantly reduced [51]. In addition to macrophages, the contents of CD4+ and CD8+ cells in aging adipose tissue increase significantly, which is more obvious in VAT [48]. Experiments have confirmed that targeted elimination of senescent cells in mouse adipose tissue can significantly improve adipose tissue dysfunction [44,52], and inhibition of SASP production by JAK/STAT inhibitors, l-carnitine and other drugs can relieve chronic inflammation and weakness in elderly mice [53,54].

Compared with WAT, BAT in high-fat diet mice often showed significantly lower immune cell enriched mRNA expression and macrophage infiltration, that is, BAT “resists” obesity induced inflammation [55]. However, continuous high-fat diet feeding can also lead to high levels of TNF-α in BAT [56], produced by the presence and increased activity of an inflammatory infiltration of immune cells. High TNF-α expression induces insulin resistance receptors in brown adipocytes, and also alters other non-insulin dependent mechanisms of BAT glucose metabolism, such as impaired FGF21-induced glucose uptake [46]. As the UCP1 activation pathway is very sensitive to pro-inflammatory signals, a high expression of inflammatory factors also impairs the thermogenic activity of BAT. Experiments have confirmed that the UCP1 level in BAT with high M1-type macrophage infiltration and high TNF-α expression level was significantly reduced [56]. Injecting a small amount of intestinal endotoxin LPS can induce low-grade inflammation in mice and reduce the expression of UCP1 in BAT [57]. TNF-α not only inhibits UCP1 production in mouse adipose tissue, but also damages UCP1 gene expression in brown adipocytes in vitro [58]. IL-1β can reduce camp-mediated UCP1 expression [59], cold-induced heat production and WAT browning in adipocytes in vivo [60].

### 3.2. Oxidative Stress

Oxidative stress plays an important role in obesity and inflammatory infiltration of adipose tissue. ROS sources are divided into the following two parts in cells: one part acts as a by-product of normal physiological processes; the other concerns certain physiological processes that intentionally produce ROS as part of signal transduction pathways or as part of cellular defense mechanisms. The first kind of ROS mainly comes from mitochondria. In the oxidative phosphorylation process, electrons always get lost, these electrons react with intracellular oxygen and produce ROS, which need antioxidant enzymes (catalase, superoxide dismutase) and related enzyme systems (such as glutathione system) to avoid damage caused by ROS. Intracellular enzymes produce ROS for different purposes, especially the NADPH oxidase NOX2 to reduce molecular oxygen, resulting in superoxide, which is used to resist pathogen attack [61].

Oxidative stress stimulates WAT deposition, changes food intake and causes obesity. From a microperspective, experiments have confirmed the involvement of ROS in adipogenic differentiation, potentiating the accumulation of intracellular lipid and increasing the size of mature adipocytes [62]. Obesity induces oxidative stress mainly through NADPH oxidase (NOX) or through dysfunctional mitochondrial oxidative phosphorylation [63,64]. In the early stages of obesity, NOX4-derived ROS from adipose cells induce insulin resistance and initiate recruitment of immune cells in adipose tissue. In the middle stage of obesity, NOX2-derived ROS produced by immune cell infiltration can aggravate adipocyte insulin resistance and adipose tissue inflammation. In late obesity, adipocyte mitochondrial ROS maintains adipose tissue inflammation and insulin resistance [64,65]. The increase of fatty acid level in obese mice activates NADPH oxidase, increases oxidative stress, reduces the activities of antioxidant enzymes SOD and GSH-Px, and leads to the dysregulation of adiponectin, plasminogen activator inhibitor-1, IL-6, MCP-1, etc. [66]. Besides, aging mice were associated with decreased adipose tissue mass and increased oxidative stress levels, which increased glutathione consumption and inhibited adipose precursor cell differentiation [67]. Experiments have shown that giving animals an antioxidant agent can help improve adipose tissue dysfunction. For example, in obese mice, treatment with NADPH oxidase inhibitors reduces ROS production in adipose tissue, alleviates adipocytokine dysregulation, and ameliorates diabetes, hyperlipidemia, and hepatic steatosis [66].

In humans, BAT is gradually lost with age. It was found that several major factors lead to age-related damage in BAT, including defects at the level of stem cells and/or progenitor cells, mitochondrial dysfunction, and impaired sympathetic tone [68]. Mitochondrial dysfunction is closely related to the continuous accumulation of ROS. According to the free radical aging theory, the increase of ROS will lead to oxidative damage of proteins and lipids, and then lead to a vicious cycle of increased production of reactive oxygen species, further aggravating oxidative damage, which may also affect the rate of mitochondrial DNA mutation in somatic cells [69]. Experiments in vitro showed that an increased oxidative stress level would damage the formation of brown adipocytes and affect the expression of heat-producing genes. Antioxidants, such as Vitamin E, can improve the phenotype of BAT oxidative stress and the change of BAT activity induced by it, and reduce SOD loss and MDA production. In addition, age-related autophagy of brown adipocytes was associated with H2O2-induced functional impairment of brown adipocytes [70]. Improving mitochondrial dysfunction caused by ROS accumulation can enhance BAT function in mice [71]. The protective effect of caloric restriction on BAT function in mice was also related to reduction of oxidative stress levels in BAT [72].

## 4. The Impact of Nutritional Interventions

Most current studies have focused on the effects of dietary supplements on adipose tissue in model animals in studies on obesity and similar. Since there is similarity between aging adipose tissue and obese adipose tissue, we combined the two for discussion.

### 4.1. Caloric Restriction

Caloric restriction (CR) is one of the main nutritional measures that has been proven to extend life span and improve age-related metabolic syndrome. Effect of CR on adipose tissue can refer to Table 1. The improvement of WAT metabolism plays an important role in the beneficial effects of CR. The research of Karl N kept the CR group intake at 73 kcal per week (16% restriction from control), and found that the CR group presented lower total mass, lean and fat mass but had a higher lean percent compared with control. The CR group showed an increase in AMPK, PGC1-α and SIRT-1, which are the downstream adiponectin signaling molecules [73]. Long-term CR was associated with high expression of FGF21, FGF21 receptor Glucose transporter 1, PPARγ coActivator-1 α and downstream targets PGC1-α and Glut1 in liver and eWAT [74]. CR has been proved to reduce VAT increase and adipocyte enlargement in elderly rats [75,76], increase hormone-sensitive lipase phosphorylation and insulin sensitivity in SAT.

CR improved the reduced WAT plasticity of middle-aged mice, decreased the levels of MCP-1 and TNF-α, improved the age-related sympathetic nerve response of BAT, and promoted the browning of WAT [16]. The browning of WAT is also one of the mechanisms by which caloric restriction reduces the total fat mass of mice, which may be related to the increased infiltration of eosinophils and M2 macrophages and the enhanced signaling of type 2 cytokines in the adipose tissues of CR animals [77]. Fabbiano S found that CR also caused changes in intestinal microbial populations. Transplantation of intestinal microbes from CR mice into germ free mice could reduce the volume and weight of adipose tissue in germ free mice and increase the expression of brown fat markers in SAT, showing the role of intestinal microbes in mediating the browning of adipose tissue. This may be related to CR reducing LPS binding to TLR4 [78]. Similar to CR, intermittent fasting results in promotion of the browning of WAT [79,80], and the mechanism might be related to changes in intestinal flora [79]. Although CR is effective in animal models, the relationship between CR and WAT browning still needs to be discussed regarding negative results in humans [80,81]. The influence of CR on the browning of subcutaneous abdominal adipose tissue was not observed in a randomized controlled trial that involved 289 overweight or obese subjects [81].

**Table 1 nutrients-14-03134-t001:** Effect of caloric restriction on adipose tissue.

Reference	Study Design	Result
Valentin Barquissau et al. [81]	289 adults (188 females and 101 males) who were overweight and ahd obesity, a dietary program with an 8 week very low calorie diet (3.3–4.2 MJ/day). Individuals achieving at least 8% of weight loss were randomized to a 6-month weight maintenance phase with ad libitum diets.	There was higher expression of brown and beige markers in women with obesity and during winter. Evolution of body fat and insulin resistance was independent of changes in brown and beige fat markers during the full intervention.
Karl N Miller et al. [73]	Male B6C3F1 hybrid mice, were randomized into control or restricted groups, fed 87 kcal week^−1^ (Bio-Serv diet #F05312), which is ~95% of ad libitum intake, or 73 kcal week^−1^, which is a 23% reduction in calorie intake from ad libitum levels and 16% reduction from controls (Bio-Serv diet #F05314).	Adipose tissue metabolism and secretory profiles change with age and are responsive to caloric restriction (CR). CR group present lower total mass, lean and fat mass but higher lean percent compared with control, Adenosine 5‘-monophosphate (AMP)-activated protein kinase (AMPK), peroxisome proliferator-activated receptor-γ coactlvator-1α (PGC1-α) and sirtuin1, which are the downstream adiponectin signaling molecule.
Namiki Fujii et al. [74]	Male 12-week-old Wistar rats, were divided into ad libitum (AL group) and calorie-restricted (CR group; 70% of AL energy intake). At 3.5, 9, or 24 months of age, rats were euthanized under isoflurane anesthesia (Mylan, Canonsburg, PA, USA).	Fibroblast growth factor-21 (FGF21) mRNA expression and/or protein increased with age in liver and WAT. Caloric restriction (CR) further upregulated FGF21 expression and eliminated the aging-associated reductions in the expression of FGFR1, beta-klotho and FGF21 target glucose transporter 1 and peroxisome proliferator-activated receptor-γ coactlvator-1α (PGC1-α) at 9 months old rat. Aging and CR upregulate FGF21 expression via different mechanisms.
Takumi Narita et al. [76]	12-weeks-old Wistar rats were divided into ad libitum (AL group) and calorie restricted (CR group; 70% of AL energy intake). Before euthanasia, CR and AL groups were further divided into fed and fasted subgroups	Caloric restriction (CR ) reduced the volume and average size of retroperitoneal white adipocytes, increased hormone-sensitive lipase (lipolytic form) phosphorylation in visceral adipose tissue and improves lipid metabolism in an insulin signaling-dependent manner in subcutaneous adipose tissue.
Salvatore Fabbiano et al. [77]	Cd45.1+ mice, Stat6−/− and Il4ra−/− mice, and their respective C57BL/6J and BALB/c controls were fed standard chow diet (16.2 MJ/kg Gross Energy; 12.8 MJ/kg Metabolizable Energy; 9 kJ% Fat, 33 kJ% Protein, 58 kJ% Carbohydrates, V1534-727, Ssniff) or 60% calorie-restricted diet (CRD) (14.6 MJ/kg Gross Energy; 7.7 MJ/kg Metabolizable Energy; 11 kJ% Fat, 51 kJ% Protein, 38 kJ% Carbohydrates, S9631-S710, Ssniff). Animals were fed daily between 18 and 19 h.	CR stimulates development of functional beige fat within the subcutaneous and visceral adipose tissue, contributing to decreased white fat and adipocyte size in lean C57BL/6 and BALB/c mice kept at room temperature or at thermoneutrality and in obese leptin-deficient mice. These metabolic changes are mediated by increased eosinophil infiltration, type 2 cytokine signaling, and M2 macrophage polarization in fat of caloric restriction animals.
Salvatore Fabbiano et al. [78]	Cd45.1+, Tlr4 KO, C57BL/6J and Germ-free (GF) mice on C57BL/6 background. Animals under caloric restriction were fed daily with 40% less standard chow diet food than the average eaten by age-matched ad libitum fed mice, and food was provided each day at 18 h.	Caloric restriction induced compositional and functional changes in the gut microbiota and promote fat browning.

### 4.2. Vitamin

Vitamins have been confirmed to be protective in age-related diseases in different literature studies, but how useful vitamins are for aging adipose tissue remains to be explored. Experiments involving rodent models, cells and humans, have been conducted and show the benefits of vitamins in adipose tissue using different pathways (Table 2).

#### 4.2.1. Vitamin A

Vitamin A and its metabolites is widely used in preventing and treating disease. Much research has focused on the role that vitamin A plays in metabolic disorders. Thanks to their good fat solubility, they usually bind to different and specific vitamin A-binding proteins or present in intracellular lipid droplets.

Vitamin A helps retain the normal physiological function of WAT and BAT. Recent studies have shown that vitamin A and retinoic acid can affect the differentiation of pre-adipocytes, the normal expression of UCP-1 and the browning of WAT [82]. Vitamin A actively affects the increase of fat content, adipocyte expansion and triglyceride accumulation caused by high fat/high fructose diet [83,84]. Feeding of high-fructose diet induced triglyceride accumulation and adipocyte hypertrophy of the visceral white adipose depots in rats. This response was not observed in vitamin A deficient groups. Vitamin A supplementation reversed the changes caused by vitamin A deficiency [83]. Vitamin A-treated rats developed higher adiposity than control rats on a high-fat diet as indicated by body composition analysis and increased WAT depot mass, adipocyte diameter, WAT DNA content, leptinemia and adipose leptin gene expression in the absence of changes in body weight [84]. High intake of vitamin A, β-carotene, and copper is significantly associated with high VAT volume [85]. The restriction of vitamin A intake increased intramuscular fat by 46% but did not affect the size of the intramuscular or subcutaneous adipocyte cells or the subcutaneous fat depth [86]. Restricted vitamin A tended to affect expression levels of Cidea and PGC-1α in scWAT, Cidea, Dio2, and increase Nfia in mesWAT, and increased expression of BMP7 and some Bmp receptors in WAT [87].

#### 4.2.2. Vitamins C and E

Vitamin C, which acts as a common antioxidant added to many processed foods, has proved to be useful in reducing inflammatory markers (hs-CRP, IL-6, and FBG) in obese patients, but causes no significant change in TC or TG [88]. Studies found that being overweight or obese was associated with low vitamin C concentration in peripheral blood. This may be related to low vitamin C intake caused by inadequate eating behavior [89]. Oral intake of vitamin C normalizes DNA methylation levels, promotes lipolysis, and decreases obesity in HFD-fed Tet1+/− mice [90]. Ascorbic acid supplementation can reduce weight gain, VAT mass and visceral adipocyte size in high-fat diet obese mice, and increased mRNA levels of PPARα and its target enzymes involved in fatty acid β-oxidation in VAT [91]. Due to the antioxidation of Vitamin C, treatment with ascorbic acid decreased ROS levels in adipose-derived stem cells derived from the omental region, and significantly improved proliferation, senescence, migration, and adipogenic capacities caused by ROS [92]. However, because vegetables and fruits are also rich in vitamins, it is difficult to accurately quantify the actual daily vitamin intake in actual population trials [93]. In some studies, vitamin C supplements were found to be ineffective [94] and even offset the beneficial effects of exercise [95].

Vitamin E participates in a variety of physiological and biochemical functions in the body, which is related to its antioxidant effect or its role as a membrane stabilizer. Alpha-tocopherol, the predominant isomer found in the body, is an efficient scavenger of lipid peroxyl radicals and is able to break peroxyl chain propagation reactions [96]. Vitamin E supplementation reduced oxidative stress, improved MDA, SOD levels of BAT in a mouse aging model [70]. There is no clear certainty about the effect of vitamin E on WAT. Vitamin E supplementation had a dramatic effect on subcutaneous fat gene expression, showing general up-regulation of significant genes [97]. Alcalá M [98] found that vitamin E could reduce oxidative stress in VAT, increase catalase, reduce glutathione peroxidase and glutathione reductase, and increase the storage capacity of adipocytes. Alcala M [99] found that short-term vitamin E supplementation was related to insulin resistance in mice on a high-fat diet. They presented a reduced generation of ROS in retroperitoneal WAT, even below physiological levels (Control: 1651.0 ± 212.0; Obese: 3113 ± 284.7; Obese and Vitamin E: 917.6 ±104.4 RFU/mg protein. C vs. OE *p* < 0.01), limited the expansion of the fat pad that increases the burden of liver lipid processing and then, as a consequence, the OE group presented as insulin resistant. While Mao G fed mice with mitochondria-targeted vitamin E derivative, and the research showed that it could reduce liver oxidative stress and inhibit fat deposition in mice [100].

Many studies have also combined vitamin E and vitamin C as antioxidants. It remains controversial whether the antioxidants supplementation reduces exercise-induced insulin resistance and mitochondrial increase. Picklo MJ proved that antioxidant supplementation blocked the ability of exercise to increase mitochondrial protein content (nicotinamide nucleotide transhydrogenase, NNT) in skeletal muscle, but did not block improvements in insulin sensitivity [101]. Rupérez FJ confirmed that vitamins decreased plasma lactic acid and thiobarbituric acid reactivity and 8-isoprostaglandin in urine in streptozotocin induced diabetic rats, and caused dramatic change in selected oxidative stress markers [102].

#### 4.2.3. Vitamin D

There are direct targets of vitamin D on adipocytes. The activated 1,25-Dihydroxyvitamin D_3_; can increase adipogenesis and decrease UCP expression. Many observational studies have found a link between reduced vitamin D levels and obesity in humans and animals, but no clear cause-and-effect relationship has been established [103]. Experiments have shown that dietary supplementation of vitamin D can regulate the function of adipose tissue. Farhangi MA found that administering Vitamin D increased the activity of the antioxidant enzymes SOD and GSH-Px in adipose tissue, decreased the concentration of TNF-α, and improved the parameters of oxidative stress and inflammation in the adipose tissue of obese rats [104]. Eugene Chang found that vitamin D deficiency was significantly associated with adipose tissue increase, macrophage infiltration and inflammation in obese rats, and proved, for the first time, that vitamin D deficiency could reduce the activities of AMPK and Sirt1 in the adipose tissue of obese rats [105]. Manna P found that vitamin D could prevent oxidative stress in the adipose tissue of obese mice and improve glucose absorption through SIRT1/AMPK/IRS1/GLUT 4 combination [106]. Wai W Cheung proved that vitamin D regulates the browning of adipose tissue and can reduce beige fat overexpression in CTNS−/− mice [107].

**Table 2 nutrients-14-03134-t002:** Effect of dietary vitamin intake on adipose tissue.

Vitamin	Reference	Study Design	Result
Vitamin A and β-carotene	Z A Kruk et al. [86].	20 female Angus steers, divided to A+(60,000 IU retinyl palmitate/100 kg/day, *n* = 10) and A-(control, *n* = 10)	The restriction of vitamin A intake increased intramuscular fat (IMF) by 46% but did not affect the size of the intramuscular or subcutaneous adipocyte cells or the subcutaneous fat depth.
	Mooli Raja Gopal Reddy et al. [83].	Male Wistar rats, divided to control (C), control with vitamin A deficiency (C-VAD), high fructose (HFr), and HFr with VAD (HFr-VAD)	Feeding of high-fructose diet induced triglyceride accumulation and adipocyte hypertrophy of the visceral white adipose depots. This response was not observed in vitamin A deficiency groups. Vitamin A supplementation reversed the changes caused by vitamin A deficiency.
	Gushchina LV et al. [108].	Wild-type (WT, C57Bl/6) and Aldh1a1(−/−) mice, divided to high-fat (HF) diet (4 IU vitA/g and 20 IU vitA/g HF diet) and control	Mild vitamin A supplementation did not influence obesity, fat distribution, and glucose tolerance in males and females of the same genotype, but multiplex analysis of bioactive proteins in blood showed moderately increased concentrations (10–15%) of inflammatory interleukin-18 and macrophage inflammatory protein-1 gamma (MIP-1γ) in vit A supplemented vs. control WT males.
	Chen HJ et al. [87].	Japanese Black steers, divided to control feed (*n* = 6) or vitamin A-restricted feed (*n* = 6) for 20 mo	The vitamin A restriction significantly increased or tended to increase expression levels of Cidea and peroxisome proliferator-activated receptor-γ coactlvator-1α (Pgc-1α) in scWAT, and Cidea, encoding D2 (Dio2), and increase Nfia in mesWAT, and increased expression of bone morphogenetic proteins (Bmp) 7 and some Bmp receptors in white adipose tissue.
	Granados N et al. [84].	Vehicle and retinyl palmitate were supplemented during the sucking period to rat pups, and were fed a normal fat or a high-fat (HF) diet for 16 weeks after weaning.	Vitamin A-treated rats developed higher adiposity than control rats on a HF diet as indicated by body composition analysis and increased WAT depot mass, adipocyte diameter, white adipose tissue DNA content, leptinemia and adipose leptin gene expression in the absence of changes in body weight.
	Arreguín A et al. [109].	21-day-old rats, supplemented during the suckling period with vehicle (controls) or an emulsion of vitamin A as retinyl ester (RE) or β-carotene (BC)	Modest vitamin A supplementation in early postnatal life impacted methylation marks in developing white adipose tissue.
	Tayyem RF et al. [85].	Face to face interview to collect data.	Participants in the highest visceral adpiose tissue volume had significantly the highest intake of vitamin A, β-carotene, and copper.
Vitamin C	Ellulu MS et al. [88].	64 obese patients, divided to experimental group (500 mg vitamin C twice a day, *n* = 33) and control group.	Vitamin C was found to have achieved clinical significance in treating effectiveness for reducing high-sensitivity C-reactive protein (hs-CRP), IL-6, and fasting blood glucose (FBG) levels, but no significant changes in total cholesterol or triglyceride were found.
	Yuan Y et al. [90].	Tet1+/+ and Tet1+/− mice are fed a high-fat diet (HFD)	Higher weight gain and more severe hepatic steatosis found in the white adipose tissue and liver of Tet1+/− mice. Oral intake of vitamin C normalized DNA methylation levels, promoted lipolysis, and decreased obesity in HFD-fed Tet1+/− mice.
	Drehmer E et al. [110].	Wistar rats were divided into four groups which received a control and hyper-lipidic liquid diet for 30 days, with or without a vitamin C supplement (CO, COC, HO and HOC)	Vitamin C did not have a protective effect on body and fat development, but vitamin C did on various metabolic parameters (glucose, pyruvate, lactate, lactate dehydrogenase, ATP, acetoacetate and beta-hydroxybutyrate) and provided positive protection against oxidative stress, especially under hyper-lipidic conditions.
Ascorbic acid	Lee H et al. [91].	C57BL/6J mice received a low-fat diet (LFD, 10% kcal fat), a high-fat diet (HFD, 45% kcal fat), or the same HFD supplemented with ascorbic acid (1% *w*/*w*) (HFD-AA) for 15 weeks.	Compared to HFD-fed obese mice, administration of HFD-AA to obese mice reduced body weight gain, visceral adipose tissue mass, and visceral adipocyte size without affecting food consumption profiles. Ascorbic acid supplementation increased the mRNA levels of peroxisome proliferators-activated receptors-α and its target enzymes involved in fatty acid β-oxidation in visceral adipose tissues.
	Sriram S et al. [93].	Adipose-derived stem cells (ASCs) were isolated from VS (omental region) and SC (abdominal region) fat depots of human subjects undergoing bariatric surgery. ASCs were also treated with vitamin C.	Treatment with Vitamin C decreased ROS levels drastically in VS-ASCs. Ascorbic acid treatment substantially improved proliferation, senescence, migration, and adipogenic capacities of compromised ASCs caused by high ROS.
Vitamin E and Vitamin C	Williams DB et al. [111].	Male Wistar rats received either a control or Vit E/C-supplemented diet (0.5 g/kg diet each of L-ascorbate and DL-all rac-alpha-tocopherol) for 9 days prior to, and during, 5 days of daily DEX treatment (subcutaneous injections 0.8 mg/g body wt)	The glucose, but not insulin, AUC was lowered with Vit E/C supplementation.
	Picklo MJ et al. [101].	Obese rat, dietary supplementation with vitamin E (0.4 g α-tocopherol acetate/kg) and vitamin C (0.5 g/kg)	Vitamin E and vitamin C supplementation with exercise elevated mtDNA content in adipose and skeletal muscle to a greater extent (20%) than exercise alone in a depot-specific manner. Vitamin E and vitamin C supplementation in obese rodents did not modify exercise-induced improvements in insulin sensitivity.
	Rupérez FJ et al. [102].	Adult female rats received streptozotocin STZ (50 mg kg^−1^) and were studied 7 or 14 days later. Rats received 5 doses of vitamins C and E over 3 days before being killed on Day 14	Adipose tissue weight progressively decreased with the time of treatment, whereas plasma triglycerides increased at 7 days, before returning to control values at 14 days after STZ treatment. Treatment with antioxidants decreased plasma lactic acid and thiobarbituric acid reacting substances, as well as urine 8-isoprostane, and decreased plasma uric acid in controls.
Vitamin E	González-Calvo L et al. [97].	7 lambs were fed a basal concentrate from weaning to slaughter (control). The other 7 lambs received basal concentrate from weaning to 4.71 ± 2.62 days and thereafter concentrate supplemented with 500 mg dl-α-tocopheryl acetate/kg (VE) during the last 33.28 ± 1.07 days before slaughter.	Vitamin E supplementation had a dramatic effect on subcutaneous fat gene expression, showing general up-regulation of significant genes, compared to CON treatment. Compared with the CON group, SAM identified a total of 330 genes with a FDR < 0.001. Among these genes, 295 were up-regulated, and 35 were down-regulated.
	Alcalá M et al. [98].	C57BL/6J mice, 28 weeks: control group (*n* = 10) fed chow diet (10% kcal from fat), obese group (O, *n* = 12) fed high-fat (HF) diet (45% kcal from fat), and obese group fed HF diet and supplemented twice a week with 150 mg of α-tocopherol (vitamin E) by oral gavage (OE, *n* = 12)	Vitamin E supplementation decreased oxidative stress and reduced collagen deposition in the VAT of OE mice, with higher vitamin E and LPO, higher catalase, lower glutathione peroxidase and glutathione reductase, allowing a further expansion of the adipocytes and increasing the storage capability.
	Alcala M et al. [99].	C57BL/6J mice were fed with a high-fat diet for 14 weeks, with (OE group) or without (O group) vitamin E supplementation (150 mg, twice per week)	O mice developed a mild degree of obesity but it was not enough to induce metabolic alterations or oxidative stress. These animals exhibited a healthy expansion of retroperitoneal white adipose tissue (rpWAT) and the liver showed no signs of lipo-toxicity while OE mice were insulin resistant although achieving a similar body weight. In the rpWAT they presented a reduced generation of ROS, even below physiological levels (C: 1651.0 ± 212.0; O: 3113 ± 284.7; OE: 917.6 ± 104.4 RFU/mg protein. C vs. OE *p* < 0.01).
	Cui X et al. [70].	Male C57BL/6J mice were divided to (a) Normal control group, (b) D-galactose (100 mg·kg^−1^) model group, and (c) D-galactose + VE (100 mg·kg^−1^·day^−1^) group at the age of 6 weeks.	Vitamin E supplementation reduced oxidative stress and improved BAT function in mouse aging model, which showed higher malondialdehyde, superoxide dismutase compared with group b.
mitochondria-targeted vitamin E derivative (MitoVit E)	Mao G et al. [100].	64 mice were fed a high-fat (HF) diet for 5 weeks. They were then switched to either a low-fat (LF) or a medium-fat (MF) diet and gavaged with MitoVit E (40 mg MitoVit E × kg body weight^−1^) or drug vehicle (10% ethanol in 0.9% NaCl solution) every other day for 5 weeks	Epididymal fat weight, as well as liver lipid and remaining carcass lipid, were significantly lowered in the MF group receiving MitoVit E (MF-E) compared to the MF group receiving vehicle only (MF-C).
Vitamin E and Vitamin D3	Lira FS et al. [112].	Male Swiss mice were divided to HFD (hyper-lipidic diet, 8 weeks, *n* = 12), HFD Vitamin D3 and HFD Vitamin E	Vitamins E and D3 supplementation were associated with lower IL-6 protein levels and the IL-6/IL-10 ratio in epididymal white adipose tissue. A 24-h treatment of vitamin D3 and vitamin E significantly reduced the interleukin-6 levels in the adipocytes culture medium without affecting interleukin-10 levels.
Vitamin D	Ionica M et al. [113].	30 consecutive patients randomized into two groups: obese (OB) (*n* = 17: 10 men and 7 women) and nonobese patients (non-OB) (*n* = 13: 7 men and 6 women) patients. VAT and vascular samples were harvested during abdominal surgery, incubated at 37 °C in endothelial cell growth basal medium (containing 0.1% BSA) in the presence or absence of the active form of vitamin D: 1,25(OH)2D3 (100 nmol/L, 12 h of incubation)	Vitamin D was able to improve the oxidative status of the white adipose tissue by mitigating the amount of total reactive oxygen species.
	Lotfi-Dizaji L et al. [114].	44 obese subjects with vitamin D deficiency (25OHD < 50 nmol/L) were assigned into vitamin D (a weight reduction diet + bolus weekly dose of 50,000 IU vitamin D) or placebo group (weight reduction diet + edible paraffin weekly) for 12 weeks.	Vitamin D supplementation resulted in significant increase of serum 25OHD level (*p* < 0.001), and significant decreased parathyroid hormone (*p* < 0.001), monocyte chemotactic protein (*p* < 0.05), interleukin-1β (*p* < 0.05) and toll-like receptor 4 (TLR4) (*p* < 0.05).
	Manna P et al. [106].	Male C57BL/6J mice were divided into control, diabetic control animals (HFD) and VD deficient (fed a VD deficient high-fat diet for 16 weeks and cholecalciferol were be gavaged to the animals at the doses 67 IU VD/kg body weight daily by oral gavage for last 8 weeks).	Vitamin D could prevent oxidative stress and upregulated glucose uptake via Sirtuin1/AMP-activated protein kinase/intracellular substrates 1/glucose transporter 4 cascade in HG-treated adipocytes and in adipose tissue of HFD diabetic mice.
	Farhangi MA et al. [104].	40 rats were divided into two groups: normal diet and high-fat diet (HFD) for 16 weeks; then, each group was subdivided into two groups including ND, ND + vitamin D, HFD, and HFD + vitamin D. Vitamin D supplementation was done for 5 weeks at 500 IU/kg dosage.	Vitamin D treatment led to a significant reduction in adipose tissue tumor necrosis factor-α and monocyte chemotactic protein-1 concentrations. Among markers of oxidative stress in adipose tissue, superoxide dismutase and glutathione peroxidase concentrations significantly increased. in adipose tissue of HFD + vitamin D treated group compared with other groups (*p* < 0.05).
	Wai W Cheung et al. [107].	LabDiet 5015 Diet (3.83 kcal/g, 3.3 IU/g vitamin D3). Ctns−/− mice and WT mice were treated with 25(OH)D3 and 1,25(OH)2D3 (75 μg/kg/day and 60 ng/kg/day, respectively) or vehicle. All mice were feed for 6 weeks. Food intake were control in two different sets.	Vitamin D supplementation could regulate excessive browning of white inguinal fat in Ctns−/− mice to a certain extent. Repletion of 25(OH)D3 and 1,25(OH)2D3 attenuated browning of beige adipocytes in Ctns−/− mice.
	Alexandra Marziou et al. [115].	High-fat diet groups were either fed with the same HFS diet or with an HFS diet supplemented with vitamin D (HFS + D; 15,000 IU·kg^−1^ cholecalciferol for 15 weeks after modeling.	VD supplementation significantly decreased monocyte chemotactic protein 1 and chemokine C-C motif ligand 5 (Ccl5) mRNA levels, which were treated as typical inflammatory markers by high-fat diet consumption.

### 4.3. Resveratrol and Other Active Substance

Polyphenols are extracted from plants and include procyanidins, catecholamines, resveratrol, and many others. Resveratrol, mainly derived from grapes and nuts, is one of the most widely used polyphenols. It was shown to be useful in reducing internal and total adipose tissue weights, activating SIRT-1, inhibiting NF-κB activation and inflammatory gene expression, and improving glucose tolerance and insulin sensitivity in animal models and humans [116,117,118]. Resveratrol also maintains BAT function and promotes the browning of WAT by activating peroxisome proliferator activating receptor signal and bone morphogenetic protein 7 (BMP7) signal, and promoting the secretion of actin and adipokine [105]. Experiment confirmed that long-term administration of resveratrol improved the adipose tissue function of rhesus monkeys fed with a long-term high-fat diet, increased the Sirt-1 protein level in VAT and inhibited NF-κB activation, reduced inflammatory factors IL-6 and IL-1β and adiponectin expression [119]. As a widely used antioxidant, resveratrol can reduce the level of ROS, improve metabolic disorder and insulin resistance and WAT inflammation [120,121,122]. Resveratrol activates AMPK, which makes sense in the operation of ROS defense system. Vitro experiments have proved that AMPK directly phosphorylates human FoxO1 (forkhead box O1) at Thr649 and increases FoxO1-dependent transcription of manganese superoxide dismutase and catalase.

In addition, other substances with antioxidant activity can also improve the function of adipose tissue. For example, walnut extract can relieve extrahepatic lipid generation, improve the level of reduced glutathione, increase the activities of GSH-Px and SOD, and reduce the accumulation of ectopic fat and its related oxidative stress state in rats [123]. Grape extract can prevent dyslipidemia by regulating lipase activity, increasing antioxidant capacity, and reducing transition metals and free radicals, such as O_2_− and OH [124]. The improvement of oxidative stress level through exercise training also has a significant effect on reducing body fat and improving metabolic syndrome [125].

In brief, WAT is redistributed while ageing, which is manifested as reduced SAT and increased VAT. In addition, increased infiltration of inflammatory cells, secretion of inflammatory factors and changes in adipocyte morphology are observed in WAT. Although rodents have fixed BAT reservoirs, aging mice show a whitening of BAT, which is characterized by inflammatory cell infiltration, decreased expression of UCP1, and a decreased number of intracellular lipid droplets and mitochondria. A large number of studies have shown that dietary intake of certain nutrients can effectively improve adipose tissue inflammation and secretory dysfunction, thereby preventing age-related metabolic diseases. Nutritional intervention is meaningful to improve adipose tissue function with different mechanisms. Caloric restriction helps retain age-related adipose function by influencing total fat mass, fat distribution and activating various biomarkers in different fat depots. Fat soluble vitamin A might cause adipose tissue accumulation. Antioxidant vitamins C and E reduce the oxidative stress in adipose tissue and are effective in weight gain and metabolic dysfunction; although the specific mechanism needs further study. Vitamin D acts directly on adipocytes, and regulates the function of adipose tissue via antioxidative and AMPK related pathway. There are also other bioactive substances, like resveratrol, which show potential effects to regulate adipose tissue function. The development of targeted substances to improve the function of aging adipose tissue is of great practical significance for healthy aging.

## 5. Conclusions

As the largest energy storage organ and endocrine organ, adipose tissue plays a key role in the occurrence and development of age-related degenerative diseases and metabolic dysfunction. Nutritional inventions are effective in adipose tissue function and have broad application prospects and far-reaching significance to reduce metabolic dysfunction, the occurrence and development of senile diseases and to realize healthy aging.

## Data Availability

Not applicable.

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
