# Peer review of "Adipose Tissue Aging and Metabolic Disorder, and the Impact of Nutritional Interventions"

_nutrients, 2022, doi:10.3390/nu14153134_

Round 1
Reviewer 1 Report
The review entitled "Senescence of Adipose Tissue, and the Impact of Nutritional Interventions" is a review of the existing studies on the effect of nutritional interventions on the senescence process in adipose tissue.
It is a well written review and recollects many studies on the field. Includes actual information and since senescence is mostly studied in cancer studies, this review may help reseachers working on metabolic diseases. There are some issues that need to be addressed.
1. The abstract needs to be re written. This section need to be a summary of the whole work (topic and findings).
2. Authors need to justify the importance of this review. Why was it necessary to perform the recollection of all the available data? They clain the importance of adipose tissue in metabolic diseases but they do not introduce the process od senescence
3. In this regard, prior to chapter 1, authors need to explain the process of senescence, in what type of studies was first considered to be important and then justify the reason to be investigated in adipose tissue. It is part of the justification of the importance of this review.
4. Authors need to include a similar table for the results of Caloric restriction in a similar way the do for the vitamin administration. In other words, it is essential to include a table of all the results. This will help other researchers to review the findings at a glance.
5. it is a bit confusing when authors use aging as secescence. It is not clear whether they consider it the same (which is not). Moreover, in many of the studies they describe they describe the impact of nutritional assessments on cell size etc. They do not mention senescence markers of at least reduction of adipogenesis markers. This is a major issue to be addressed. Otherwise the title of the review needs to be changed.
6. Conclusions need to be more specific and provide a take home message. Are nutritional interventions adequate? which one o which group of these interventions is promising etc
Reviewer 2 Report
The authors present work related to the senescence of adipose tissue, and the impact of nutritional interventions. The work has potential, however, the entire manuscript has significant problems with the English grammar. The text must be extensively revised for the English writing before further review.
Author Response
Thank you for your comment. We have modified the language of the full text.
Round 2
Reviewer 1 Report
The study of Xiujuan Wang, Meihong Xu and Yong Li present a review of different studies focused in physiological changes ocurring in different adipose tissues due to aging/metabolic pathologies especially obesity.
The revised version of the manuscript is improved and more complete than the initial version. However, the main concern of the manuscript is that even though authors begin to mention the effects of aging, the major part of the review describes the effects of obesity or metabolic syndrome in the physiology of adipose tissues. It is true that metabolic pathologies accelerate the aging of this tissue and they share metabolic pathways, the title of the manuscript is not representative of the manuscript.
The section of nutritional intervention includes metabolic pathologies and not aging alone. Authors need to separate studies including humans and animal models.
The format of the tables need to be improved as well the description of the studies and their results.
Author Response
Thank you for your comments on the above manuscript and for the opportunity to revise it. We have taken the Editor’s comments and suggestions into careful consideration and revised the manuscript accordingly. On the following pages, please find our point-to-point responses to the Editor’s concerns in the order that they were originally listed, and details of the pages on which the changes have been made. Point 1: ……the major part of the review describes the effects of obesity or metabolic syndrome in the physiology of adipose tissues. It is true that metabolic pathologies accelerate the aging of this tissue and they share metabolic pathways, the title of the manuscript is not representative of the manuscript. Response 1 : Thank you for your comment. We change the title as " Adipose tissue aging and metabolic disorder, and the impact of nutritional interventions" Point 2 and 3:The section of nutritional intervention includes metabolic pathologies and not aging alone. Authors need to separate studies including humans and animal models. The format of the tables need to be improved as well the description of the studies and their results. Response 2 and 3: Thank you for your comment. We adjusted the order of the table, and the experiments related to people are now in front of the table. The description in the table were revised.